# Efficacy of Rapid Maxillary Expansion with or without Previous Adenotonsillectomy for Pediatric Obstructive Sleep Apnea Syndrome Based on Polysomnographic Data: A Systematic Review and Meta-Analysis

**Vincenzo Quinzi [1]**, **Sabina Saccomanno [1],\***, **Rebecca Jewel Manenti [1]**, **Silvia Giancaspro [1]**, **Licia Coceani Paskay [2]** and **Giuseppe Marzo [1]**

1. Department of Health, Life and Environmental Science, University of L'Aquila, Piazza Salvatore Tommasi, 67100 L'Aquila, Italy; vincenzo.quinzi@univaq.it (V.Q.); rebeccajewel.manenti@student.univaq.it (R.J.M.); silvia.giancaspro@student.univaq.it (S.G.); giuseppe.marzo@univaq.it (G.M.)
2. Academy of Orofacial Myofunctional Therapy (AOMT), 910 Via De La Paz, Ste. 106, Pacific Palisades, CA 90272, USA; lcpaskay@aomtinfo.org
* Correspondence: sabinasaccomanno@hotmail.it

**Abstract:** Aim: To systematically review international literature related to rapid maxillary expansion (RME) as the treatment for obstructive sleep apnea syndrome (OSAS) in children less than 18 years-old, followed by a meta-analysis of the apnea-hypopnea index (AHI) before and after RME, with or without a previous adenotonsillectomy (AT). Methods: Literature on databases from PubMed, Wiley online library, Cochrane Clinical Trials Register, Springer link, and Science Direct were analyzed up to March 2020. Two independent reviewers (S.G. and R.J.M.) screened, assessed, and extracted the quality of the publications. A meta-analysis was performed to compare AHI values before and after the treatment with RME. Results: Six studies reported outcomes for 102 children with a narrow maxillary arch suffering from OSAS with a mean age of 6.7 ± 1.3. AHI improved from a M ± SD of 7.5 ± 3.2/h to 2.5 ± 2.6/h. A higher AHI change in patients with no tonsils (83.4%) and small tonsils (97.7%) was detected when compared to children with large tonsils (56.4%). Data was analyzed based on a follow-up duration of ≤3 year in 79 children and >3 years in 23 children. Conclusion: Reduction in the AHI was detected in all 102 children with OSAS that underwent RME treatment, with or without an adenotonsillectomy. Additionally, a larger reduction in the AHI was observed in children with small tonsils or no tonsils. A general improvement on the daytime and nighttime symptoms of OSAS after RME therapy was noted in all the studies, demonstrating the efficacy of this therapy.

**Keywords:** obstructive sleep apnea syndrome; rapid maxillary expansion; apnea-hypopnea index; polysomnography; adenotonsillectomy

---

## 1. Introduction

Obstructive sleep apnea syndrome (OSAS) is defined as the extreme end of the spectrum of obstructive sleep disordered breathing in children with a protracted partial upper airway obstruction (hypopnea) and/or an intermittent complete obstruction (apnea) [1,2]. This disorder presents repeated events of partial and complete airway obstruction during sleep that brings to a disruption of normal ventilation, decrease in oxygen saturation, arousals and more severe impairments in cognitive function [1,3]. It is associated with sleep loss and intermittent hypoxia [3]. Childhood OSAS is an important risk factor for childhood developmental disorders, metabolic disorders and inflammation [4].

It is common in children with a prevalence ranging between 1.2 and 5.8% of the general pediatric population (<18 years old) [2]. Although the etiopathogenesis of this disorder in adults is known, many characteristics of this syndrome in children are debated. OSAS and is known to be highly connected to the size of the upper airways and adenoid tissue [5] OSAS can and have different levels of severity that could give rise to long term effects on children including alterations in behavior and neurocognitive deficits affecting attention, learning and memory as well as executive and motor functions [1,3]. According to the American Academy of Sleep Medicine (AASM) both apneas and hypopneas are observed in this syndrome. Apnea is known as the interruption of airflow for 10 s or more, while hypopnea is a reduction of >30% in air flow for a period lasting at least 10 s associated with a >3% decrease in oxygen saturation or arousals [6]. According to the AASM the diagnosis is based on both nocturnal and daytime symptoms and a well-documented full-night polysomnography [7]. For an OSAS diagnosis a threshold of 15 events per hour of recording is applied. The apnea-hypopnea index (AHI) that calculates the sum of the apneas and hypopneas per hour of sleep defines the severity of OSAS. It can be classified as a mild OSAS (AHI = 1 to 4.9), moderate OSAS (AHI = 5 to 9.9) or severe OSAS (AHI > 10) [6–8]. While obesity is considered to be a major cause of OSAS in adults, adenotonsillar hypertrophy (ATH) which may increase airways resistance, is thought to be the prime cause of childhood OSAS [4,8,9], as it contributes to the narrowing of the retro-palatal area that already has the smallest cross-sectional area and therefore is the prevalent site of obstruction [5,9–11]. In literature it is seen that ATH leads not only to problems related to mouth breathing, snoring, chronic sinusitis, nasal congestion, hyponasal speech, but also to emotional disorders and poor neurological development [7]. When fat is present in the area of the pharyngeal soft tissue it decreases the caliber of the lumen and it also increases the collapse of the structures [5,9,11]. Additionally, the increased occurrence of fat in the thoracic and abdominal walls importantly reduces the respiratory function in these patients [5]. In fact, obese and overweight children compared to normal-weight children present a higher risk of developing OSAS [5,10,12]. However, the primary structural abnormality present in obese children with OSAS is the lymphoid tissue, rather than other soft tissue components [9]. Obesity features, long face syndrome, elevated systemic blood pressure and craniofacial alterations characterize the clinical presentation of children with OSAS. During the physical examination it is possible to find adenotonsillar hypertrophy [5,9,13], inflammation of the nasal mucosa, deviation of the nasal septum, pseudo-macroglossia, narrow upper airway, maxillary constriction and/or some degree of mandibular retrusion [5,13,14]. The dento-facial orthopedic treatment procedure of rapid maxillary expansion (RME) is routinely used in young patients to treat constricted maxillary arches [15,16] and it is also considered a potential additional treatment in children with OSAS. Through methods of facial analysis, it is common to find an increase of the craniomandibular and intermaxillary angles (high angle face) and a posterior rotation of the mandible in children with OSAS [17,18]. The obstructive sleep apnea symptoms in children are divided into nocturnal and diurnal, with important neurobehavioral effects [1,5,19]. Among the diurnal, the most prevalent symptoms are daytime sleepiness, frequent headaches, hyperactivity, nasal speech [5], rebelliousness, inattentive behavior [1,20,21], depression [1,5,22], mood instability, irritability, and aggressiveness [1,4,6,20,23]. Nocturnal symptoms include snoring, excessive sweating, observed apnea, intermittent hypoxemia, hypercapnia, sleep fragmentation [5,6,24], gasping, oral breathing, sleepwalking and nightmares, paradoxical thoracic movements, and nocturnal enuresis [6,25]. One of the main reasons why this diagnosis is relevant especially in teenagers is because of the association between a poor academic performance and sleep disordered breathing and daytime sleepiness [4,23]. Furthermore, children affected by severe OSAS may develop early metabolic syndromes: obesity, dyslipidemia, insulin resistance as well as systemic hypertension [5]. Particular attention should be given to the presence of a short lingual frenulum, which is also associated with difficulties in sucking, swallowing and speech. The presence of this anatomical anomaly induces oral dysfunctions that can lead to oro-facial dysmorphism that decreases the size of the upper airway increasing the risk of upper airway collapsibility during sleep [26,27]. The precise role that genetics plays in the pathogenesis of pediatric

OSAS is still a matter of debate, but there are some clinical syndromes such as Down syndrome, midface hypoplasia and neuromuscular disorders [5,28] that increase the chances of developing OSAS. This underlines the fact that any acquired or congenital condition involving a respiratory control center may potentially lead to the development of OSAS [5]. Some inflammatory factors and biomarkers like inflammatory cytokines are related to pediatric OSAS [28,29]. Furthermore, it has been shown that allergic rhinitis can affect sleep because nasal congestion secondary to a nasal mucosa inflammatory process induces an increase in the airway resistance that could result in oral breathing and sleep disruption [5]. Childhood OSAS is difficult to diagnose because, in contrast to adults, children do not usually complain of daytime sleepiness, which is considered to be one of the most significant neurological consequences [1]. The gold standard test to diagnose childhood OSAS is an in-laboratory polysomnography (PSG) [30]. This diagnostic procedure gives various quantitative parameters to assess the respiratory function such as the lowest oxygen saturation (LSAT) and apnea-hypopnea index (AHI) [30,31]. Patients that present a narrow and/or high arched hard palate are more prone to the development of OSAS and often present dental malocclusions that can be treated with RME [32]. The main goal of RME is to widen the maxillary bone and the maxillary dental arches, to solve existing posterior crossbites [33] in order to reduce maxillary constriction and mouth breathing, thereby helping solve nasal-respiratory problems. This study was designed following the PICO guidelines and its objective was to update the knowledge of the treatment of pediatric OSAS through literature reviews describing main interventions and management strategies. Specifically, this study systematically reviewed international literature on RME used as a treatment of OSAS in children < 18 years old, followed by a meta-analysis on the polysomnographic data before and after RME, with or without previous adenotonsillectomy (AT). The AHI index was analyzed in this meta-analysis to quantify the pre and post RME polysomnographic data. A sub-analysis was made in order to find the AHI outcomes based on prior AT and tonsil size before the therapy.

## 2. Material and Methods

### 2.1. Guidelines

The PRISMA (Preferred Reporting Items for Systematic Reviews and Meta-Analysis) statement was adhered to as much as achievable.

### 2.2. Search Strategy

The following electronic databases were searched up to March 2020. A bibliographic review limited to humans was conducted, characterized by a publication date range from 2004 to 2020 with the following electronic databases: Medline full text, PubMed, Cochrane Clinical Trials Register, PEDro, and Science Direct. Literature review was summarized in Table 1. Search terms in the databases previous mentioned were "diagnosis" AND "pediatric" AND "sleep" AND "apnea" OR "OSAS" AND "maxillary expansion" OR "RME" OR "RPE" AND "pediatric OSAS" OR "Apnea-hypopnea index" OR "AHI values" AND "childhood OSAS" OR "adenotonsillectomy" OR "AT" AND "pediatric OSAS" OR "mandibular advancement device" OR "MAD" AND "pediatric OSAS" OR "pediatric OSAS"OR "orofacial myofunctional therapy" OR "OMT" AND "pediatric OSAS." To avoid inappropriate exclusion, abbreviations of all keywords were used. Restriction in language was not applied. A manual search of the references of chosen studies was performed. SCImago Journal ranking (SJR) was used to measure the scientific influence of the cited articles' journals chosen to write this paper. Accordingly, the journals from where the main articles originated, showed a high impact factor (Q1-Q2) with highly cited researchers according to their Google Scholar public profiles.

The protocol of this research was registered in the publicly accessible PROSPERO (CRD42020170909), the primary database for registering systematic review protocols.

**Table 1.** Literature review (tabular form).

| Author | Investigator | Title | Source | Findings of the Study |
|---|---|---|---|---|
| Villa et al. 2011 | [13] | Efficacy of rapid maxillary expansion in children with obstructive sleep apnea syndrome: 36 months of follow-up | Sleep and Breathing | After RME treatment, the AHI decreased, and the clinical symptoms had resolved. |
| Marino et al. 2012 | [17] | Rapid maxillary expansion in children with Obstructive Sleep Apnoea Syndrome (OSAS) | European Journal of Pediatric Dentistry | The nasopharyngeal airway measurements showed a significant increase after treatment with RME. |
| Fastuca et al. 2015 | [31] | Airway compartments volume and oxygen saturation changes after rapid maxillary expansion: A longitudinal correlation study | Angle Orthodontist | Oxygen saturation was increased, and the apnea/hypopnea index was improved. |
| Pirelli et al. 2015 | [34] | Rapid maxillary expansion (RME) for pediatric obstructive sleep apnea: a 12-year follow-up | Sleep Medicine | PSG showed an improvement in the AHI value and oxygen saturation nadir. Moreover, a total resolution of clinical complaints was reported. |
| Guilleminault et al. 2011 | [35] | Adeno-tonsillectomy and rapid maxillary distraction in pre-pubertal children, a pilot study | Sleep Breath | Children presented an improvement of both clinical symptoms and PSG findings. Nevertheless, none of the children presented normal results after treatment 1 (only RME or Adenotonsillectomy). In fact, both treatments are needed to obtain normal results. |
| Kim 2014 | [36] | Orthodontic Treatment with Rapid Maxillary Expansion for Treating a Boy with Severe Obstructive Sleep Apnea | Sleep Medicine | Banded RME was used to correct the quality of sleep and improve the narrow maxillary arch. |

### 2.3. Study Selection

The selection process of this study was done in two phases. In the first, the studies were considered according to the following inclusion criteria (A):

- All studies designs were investigated
- Patients who were treated with rapid maxillary expansion appliances
- Patients demonstrating a narrow hard palate and/or a high arched hard palate
- Presence of polysomnographic data related to the AHI index before and after RME therapy
- Studies involving children and adolescents < 18 years old with OSAS
- All languages
- Any publication years

The studies accepted to the second phase where only the ones that satisfied all of the inclusion criteria, which were expressed in an investigation of the preselected studies according to the following exclusion criteria (B):

- OSAS was considered if the obstructive apnea-hypopnea index (AHI) was ≥2/h
- Studies that didn't discuss RME as treatment for OSAS were excluded
- Studies that lack to provide quantitative data were excluded

### 2.4. Data Screening and Extraction

Two reviewers (S.G. and R.J.M.) independently performed screening. Disagreement regarding inclusion was resolved by discussion. During the initial screening, article's title and abstract, when available, were investigated. The articles were selected if considered relevant by at least one reviewer. A detailed full-text analysis was performed, and two reviewers extracted data from each study such as:

main and secondary outcomes, study purpose, design and participants. Dissentions related with data screening and extraction were solved by accord between the two reviewers.

### 2.5. Outcome Measures

The primary objective was to evaluate the pre and post-RME apnea-hypopnea index with or without previous surgery. As a secondary objective was to evaluate the pre and post-RME AHI index outcomes were assessed, based on tonsil size and if the patient was prior treated with AT or not. Additionally, daytime and nighttime symptoms before and after rapid expansion were studied.

### 2.6. Statistical Analysis

The decision to carry on a meta-analysis was made if sufficient parallelisms between studies in types of participants, analyzed outcomes and interventions were present. For the reproducibility of this study, statistical analysis was described. This study was made collecting different data from the medical literature, which were analyzed and reworked. The final results were reported as M ± SD, where M is a weighted mean and SD is the standard deviation. The weighted mean was calculated using only one weight (number of patients) for AHI change percentage and two weights (number of patients and SD) for the other parameters. SD was calculated as the difference between M and the minimum weighted mean that was evaluated considering each value reduced of SD. AHI change percentage was estimated as a kind of error: $|(\text{Pre AHI-Post AHI})/\text{Pre AHI}| \times 100$. Review Manager (RevMan) Version 5.3 (Copenhagen: The Nordic Cochrane Center, The Cochrane Collaboration, 2014) was employed for this meta-analysis.

Due to scarcity of randomized clinical trials on the treatment of childhood OSAS, 46 articles were also included to qualitatively explore this lower quality evidence regarding treatment.

## 3. Results

### Study Selection

At the beginning, according to the electronic and manual search, 4129 articles were found (see Table 2).

**Table 2.** Abstracts Retrieved by Electronic, Manual and Reference Searching.

| Search Method No. of Abstracts without Overlap | |
| --- | --- |
| Wiley Online | 1200 |
| PubMed | 611 |
| Cochrane Controlled Clinical Trials Register | 8 |
| Springer Link | 1435 |
| Science Direct | 855 |
| Reference selected articles | 20 |
| Total | 4129 |

A total of 4129 published articles were considered from the initial search. After excluding duplicates (3925) and unsuitable studies (150), 52 studies were included. Figure 1 exhibits a flowchart of the search process used in this review that is based on PRISMA guidelines for systematic reviews.

General Characteristics of this Study. The decision of analyzing only AHI values in the polysomnographic data was done in order to standardize the articles with the only common outcomes observed in all the 102 children of this meta-analysis. In fact, Pirelli et al. [34], Fastuca et al. [31], and Guilleminault et al. [35] have analyzed the outcomes of AHI and LSAT indexes. Villa et al. [13] have reported outcomes of AHI index and mean oxygen saturation index (MSAT). Kim's case report [36] analyzed AHI, LSAT and respiratory disturbance index (RDI) outcomes. Marino et al. [17] described cephalograms and AHI (see Table 3).

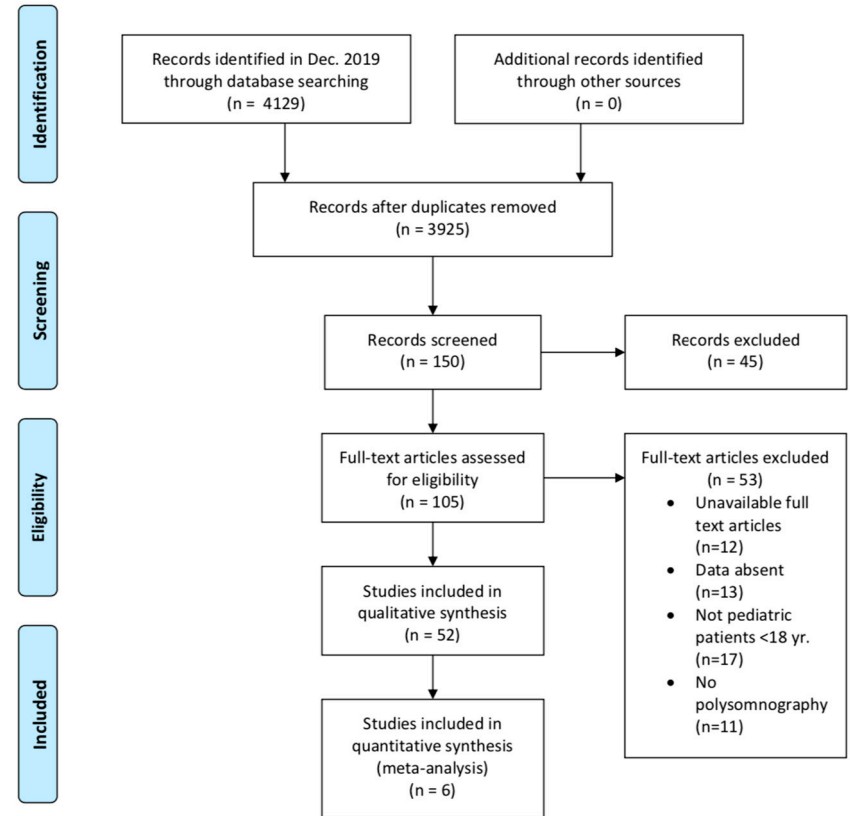

**Figure 1.** Study search flowchart.

**Table 3.** General Characteristics of Included Articles.

| Study, Design | Study Site | Outcomes Analyzed |
|---|---|---|
| Pirelli et al. 2015, PCS | Italy | AHI, LSAT |
| Fastuca et al. 2015, PCS | Italy | AHI, LSAT |
| Kim 2014, RCR | Korea | AHI, LSAT, RDI |
| Marino et al. 2012, RCS | Italy | AHI, CEPH |
| Guilleminault et al. 2011, PRT | France-Italy | AHI, LSAT |
| Villa et al. 2011, PCS | Italy | AHI, MSAT |

AHI = apnea–hypopnea index; CEPH = cephalograms; LSAT = lowest oxygen saturation; MSAT = mean oxygen saturation; RDI = respiratory disturbance index in supine position and in lateral position; PCS = prospective case series; PRT = prospective randomized trial; RCR = retrospective case report; RCS = retrospective case series.

Primary outcome. The demographic and AHI values pre and post rapid maxillary expansion, with or without surgery in children < 18 years old were summarized in Table 4.

**Table 4.** Demographic and Polysomnographic Data Before and After Rapid Maxillary Expansion with or without Surgery in Children.

| Study | No. | Age, Year * | BMI, kg/m² | F/U | Pre-RME AHI | Post-RME AHI | AHI% Change |
|---|---|---|---|---|---|---|---|
| Pirelli et al. 2015 | 23 | 8.6 | 22.7 ± 1.3 | 12.3 ± 1.5 year | 12.2 ± 2.6 | 0.4 ± 1.6 | −97.7% |
| Fastuca et al. 2015 | 22 | 8.3 ± 0.9 | NR | 1 year | 5.0 ± 1.5 | 1.5 ± 0.6 | −70% |
| Kim 2014 | 1 | 11 | 22.4 | 2 year 5 month | 18.9 | 1 | −94.7% |
| Marino et al. 2012 | 15 | 5.9 ± 1.6 | NR | 1.6 ± 0.6 year | 4.5 ± 3.8 | 3.4 ± 4.3 | −24.4% |
| Guilleminault et al. 2011 | 31 | 6.5 ± 1.1 | NR | 3 month | 7.9 ± 3.2 | 3.1 ± 2.3 | −60.7% |
| Villa et al. 2011 | 10 | 6.6 ± 2.1 | 16.7 ± 3.6 | 2 year 11 month | 6.3 ± 4.7 | 2.3 ± 1.7 | −63.4% |
| TOTAL | 102 | 6.7 ± 1.3 | 19.4 ± 2.5 | ≤3 year | 7.5 ± 3.2 | 2.5 ± 2.6 | −66.1% |

* Age at baseline; * Mean oxygen saturation (MSAT); AHI = apnea–hypopnea index; BMI = body mass index at baseline; F/U = follow-up; NR = not reported; RME = rapid maxillary expansion.

Apnea-Hypopnea index values. From the pre-RME to the post-RME, the overall AHI decreased by 66.1%. In fact, AHI improved from a M ± SD of 7.5 ± 3.2/h to 2.5 ± 2.6/h. Regarding AHI < 1/h, no success rates were reported in the studies by Pirelli et al., Villa et al., Guilleminault et al., Marino et al., and Fastuca et al. Villa et al. as well as the other studies, demonstrated a decrease in the AHI index 24 months after the end of the orthodontic treatment. Guilleminault et al. demonstrated a success rate only after "Treatment 2". The 31 children in their study were divided into two groups: group 1 received surgery followed by orthodontics, while group 2 received orthodontics followed by surgery. After "Treatment 1", the AHI values in group 1 decreased from 12.5 ± 0.8/h to 4.9 ± 0.6/h. After "Treatment 2", the AHI values in group 1 decreased to 0.9 ± 0.3. Instead, after "Treatment 1" the AHI values in group 2, decreased from 11.1 ± 0.7/h to 5.4 ± 0.6/h. After "Treatment 2", the AHI values in group 2 decreased to 0.9 ± 0.3 same as group 1, obtaining an almost normalized level after the two treatment modalities. The case study carried out by Kim showed a decrease in AHI from 18.9/h to 1.0/h two years and five months after orthodontic treatment. The random effects model calculation (n = 102 patients) demonstrate a mean difference of 5.11/h (95% CI = 4.58 to 5.64), overall effect z = 18.96, and $p$ = 0.00001. The Q statistic is $p < 0.00001$ (significant heterogeneity) and $I^2$ = 97% (high inconsistency; see Figure 2). We acknowledged the high heterogeneity either within the study population and between the studies, so in order to take in account the most influent factor we choose fixed effect model that emphasize within study weight. We provided also random effect models to not compress between study variance.

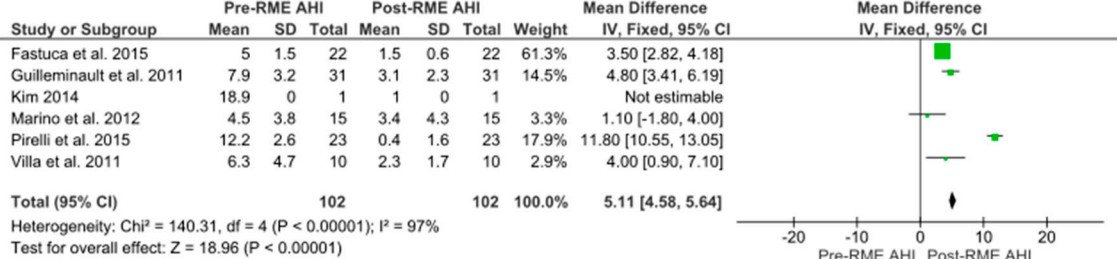

**Figure 2.** Apnea-hypopnea index (AHI) mean difference and standardized mean difference before and after rapid maxillary expansion (RME). CI = confidence interval; IV = inverse variance; SD = standard deviation.

Lowest Oxygen Saturation. Marino et al. did not report data about LSAT while mean oxygen saturation was provided by Villa et al.

All the studies except for Marino et al., in which it was not reported, demonstrated an improvement of the lowest oxygen saturation level.

Secondary Outcome. A sub-analysis was performed to understand if there were any differences between children who underwent previous AT and children who did not undergo AT (see Table 5)

**Table 5.** Pre and Post RME Outcomes Based on Prior Adenotonsillectomy and Tonsil Size.

| Tonsil Status | Pre-RME AHI | Post-RME AHI | AHI% Change |
|---|---|---|---|
| No tonsils, n = 17 | 4.9 ± 0.6 | 0.9 ± 0.3 | −82.4% |
| Small tonsils, n = 23 | 12.2 ± 2.6 | 0.4 ± 1.6 | −97.7% |
| Large tonsils, n = 24 | 7.1 ± 4 | 3.3 ± 1.3 | −56.4% |

Not reported:

- Marino et al. and Fastuca et al. did not report either tonsil sizes or whether any previous surgery was performed. Thus, these studies were excluded from the sub-analysis. Therefore, this sub-analysis was performed on 65 out of 102 children.

Previous AT:

−   Kim's case report subject was treated previously with AT, but the child did not respond to the treatment, done three years before the RME.
−   Guilleminault et al. investigated in group 1 the RME after AT in 16 children with narrow maxilla and with a narrow and high hard palate; their AHI improved from 4.9 ± 0.6/h to 0.9 ± 0.3/h.

No prior surgery:

−   Pirelli et al. noted that their patients didn't undergo any prior surgery (small tonsils).
−   Villa et al. excluded those previously treated with AT. They excluded children that had a history of prior OSAS therapies (including tonsillectomy and adenoidectomy). Before the placement of the RME device, all children underwent an otorhinolaryngological examination to grade their tonsillar hypertrophy (enlarged palatal tonsils) following a standardized scale ranging from 0 to 4 (large tonsils). Their study reported children presenting tonsillar hypertrophy as grade 2 (3 out of 10), grade 3 (5 out of 10), and grade 4 (2 out of 10).
−   Guilleminault et al. the used RME appliance in 14 children (group 2) that presented a grade 2 or larger tonsils and a narrow or high hard palate and a narrow maxilla, showing an improvement in AHI from 11.1 ± 0.7/h up to 5.4 ± 0.6/h in children with large tonsils.

Comorbidities. The majority of the studies mentioned the exclusion of patients with comorbidities and syndromes.

## 4. Discussion

The main findings of this article rely on the AHI values pre and post RME with or without previous AT.

Related to the primary outcome, the AHI in 102 children with a mean age of 6.7 ± 1.3 with obstructive sleep apnea decreased after RME treatment. The AHI improved from a M ± SD of 7.5 ± 3.2/h to 2.5 ± of 2.6/h. A 66.1% improvement in AHI was detected. The most successful group based on percentage reduction of the AHI were the 23 children in the study of Pirelli et al. [34] with a 97.7% compared to the least successful where the percentage was 24.4% (Marino et al.) [17]. Aside from this study, RME demonstrated the capacity to provide at least a 60.7% reduction in the AHI value.

In the meta-analysis by Camacho et al. [32], similar outcomes were described: AHI in their 313 patients demonstrated a 70% improvement. The pre RME apnea-hypopnea index was 7.5 ± of 3.2/h, which is considered by the ASSM to be a mild-to-moderate OSAS. A severe OSA was identified before RME therapy on Kim's case report [36], with an AHI value of 18.9/h and on Pirelli et al. [34] with an AHI value of 12.2 ± 2.6/h. Moderate-to-severe OSAS were observed by Villa et al. [13] and by Guilleminault et al. [35] while a moderate OSA was analyzed in the studies of Fastuca et al. [31] and Marino et al. [17].

Regarding the secondary outcome, to evaluate the pre and post-RME AHI index outcomes based on tonsil size and if the patient underwent a prior AT or not, it was possible to observed a higher AHI change in patients with no tonsils (83.4%) and small tonsils (97.7%) in comparison with children characterized by large tonsils (56.4%). Similar outcomes were examined in the study of Camacho et al. [32] that indicated a higher AHI change in patients with no tonsils (85%) and small tonsils (95%) when compared to children with large tonsils (61%). In fact, in a sample of 33 children with large tonsils their AHI decreased from a M ± SD of 11.4 ± 11.6/h to 4.5 ± 3.6/h after RPE. The authors of this study concluded that patients with large tonsils might continue to exhibit oropharyngeal obstruction although there are improvements at the level of the palate, underlying the importance of preceding the RPE with AT. Guilleminault et al. [35] stated that none of the children presented normal results after treatment 1 (only surgery or only orthodontics), with the exception of one case. Thirty children underwent treatment 2 with an overall significant improvement. This confirms the importance of an interdisciplinary approach on the treatment of children with OSAS, since both RME and AT are needed treatments. Nevertheless, there was no significant difference in the amount of improvement

noted independently of the first treatment approach. Kim's case report [36] reported as well that performing exclusively AT is not sufficient to treat OSAS. Failure of RME treatment alone was observed in the study of Villa et al. [13] only in an 8-year-old girl in whom the AHI did not change after 12 months of RME treatment, but instead increased after 24 months. The authors reported that she had tonsil hypertrophy, which is congruent with the secondary outcome of this meta-analysis. Related to the risk of bias, it may lie on the choice of literatures. Kim's study is a case report, and some others are included the literature from same authors. Nevertheless, this analysis demonstrated a general reduction of the daytime and nighttime symptoms of OSAS after RME therapy with or without previous AT. An overall improvement in the AHI index and in the symptoms related to sleep-disordered breathing was seen in the six articles used for this metanalysis. In fact, in Pirelli et al. prospective case series [34], 23 children were evaluated with clinical interviews and evaluation, completion of the Pediatric Daytime Sleepiness Scale (PDSS) or Epworth Sleepiness Scale (ESS) and the Italian translation of the Pediatric Sleep Questionnaire. Every child underwent their annual orthodontic visits until their final follow-up which based on subjective reports through questionnaires filled by the parents showed improved scores and absence of symptoms related to sleep disordered breathing. Stable clinical improvements were seen through CT imaging by comparing the base width of the maxilla pre and post treatment with RPE. In the study by Villa et al. [13] a stable reduction in the clinical and polysomnographic signs and symptoms of OSAS was observed in the majority of the children treated.

In Kim's case report [36] sleep architecture showed a general improvement after the treatment of RME in a child that didn't respond to previous AT. It remains unclear though whether RPE just enlarges the bony framework that defines the nasal passages, or additionally, enlarges the oropharynx (soft tissue). In the study by Guilleminault et al. [35], both groups underwent a night of nocturnal recording at entry and after each treatment. Respiration was always monitored, and the overall clinical symptoms improved. According to the study by Fastuca et al. [31], after the end of the treatment the upper, middle and lower airway compartments showed an important volume increase. Furthermore, according to the findings by Marino et al. [17], 52% of the preschool children analyzed in the study showed a decrease in the respiratory disturbance index (RDI) by more than 50% after the RME treatment, though the sample size was small. Although RME treatment for OSAS showed overall positive results based on the articles of this meta-analysis it may give negative or neutral results. As previously mentioned, it remains unclear if the RME appliance has an enlarging action also on the soft tissue other than the bony framework. According to the American Academy of Pediatrics, for children diagnosed with OSAS and showing clinical signs of adenotonsillar hypertrophy but without any counter-indication to surgery, the recommended first line of treatment was AT. However, in the OSAS treatment, since residual lymphoid tissue may contribute to a persistent obstruction, it is very important to remove both the adenoid and the tonsillar tissue [37]. Although subtotal resection also carries an increased risk of tonsillar regrowth [8], tonsillar tissue reacts to pathogens such as bacteria, dust, or pollen in the body, so partial removal of tonsillar tissue without identification and removal or reduction of pathogens will not be successful neither in the short nor in the long run [38]. Regarding the Quality of Life (QoL) index, ADT has a positive impact on the overall QoL of pediatric patients [39,40]. Although adenotonsillectomy is considered an effective treatment, recent studies have demonstrated that even if the majority of children showed an improvement in their PSG parameters post-surgery, not all children achieved a total normalization, while in healthy, non-obese children, the success rate of adenotonsillectomy is approximately 75% [8]. Another orthodontic appliance used in pediatric patients to treat OSAS is the mandibular advancement device (MAD) [41]. The appliance mechanically protrudes the mandible, thus moving the tongue forward, with the aim of preventing collapse of the upper airway [42]. Anteroposterior (AP) mandibular protrusion increases the airway caliber at the retropalatal area via lateral expansion and displaces parapharyngeal fat pads while the tongue and tongue-base muscles move forwards. The airway improvements can be appreciated with magnetic resonance imaging and cone-beam computed tomography as well as nasal endoscopy [43]. Additionally, it is important to mention another option: orofacial myofunctional therapy (OMT), which

was found efficacious in treating childhood OSAS. The function of this therapy is to promote changes in the upper airways' muscles. The AHI before and after OMT reported by the articles reviewed in this study decreased and the ΔAHI%: (AHI before − AHI after)/AHI before × 100) [44] was 38.8% in Guimarães et al. [45], 48.2% in Baz et al. [46], 50.4% in Diafería et al. [47,48], 28.4% in Ieto et al. [49], 58.0% in Villa et al. [50], 2.0% in Verma et al. [51], and 22.5% in Mohamed et al. [52]. It was found that OMT was indicated to treat residual OSA in children after surgical treatment. OMT is based on exercises and other strategies that are intended to improve sensitivity, proprioception, mobility, coordination and strength of orofacial structures. These exercises are also intended to promote a good performance of respiration, mastication, deglutition and speech [51,52]. According to Villa et al. OSAS does not resolve following first-line treatment in a considerable number of children, so oropharyngeal exercises may be contemplated as a complementary therapy to AT to effectively treat pediatric OSA [52]. Regarding the follow-up period, data was analyzed based on a follow-up duration of ≤3 year in 79 children and >3 years in 23 children. In fact, only Pirelli et al. [34] analyzed data on a follow-up of 12.3 ± 1.5 years. Thus, a future recommendation could be to investigate the efficacy of RME in the long-term treatment of OSAS.

## 5. Conclusions

On a mean follow-up duration of ≤3 years, a decrease of 66.1% of AHI was detected in all of the 102 children with OSAS that underwent RME treatment, with or without AT. A larger AHI reduction was observed in children with small tonsils (97.7%) or no tonsils (82.4%) rather than large tonsils (56.4%). This data highlighted the importance of AT combined with RME treatment. Furthermore, a general improvement of the daytime and nighttime symptoms of OSAS after RME therapy was documented in all the studies considered, demonstrating the efficacy of this therapy.

**Author Contributions:** G.M. was the coordinator, S.S. and V.Q. were the principal investigators, S.G. and R.J.M. contributed in writing the manuscript and L.C.P. contributed in both editing and writing. All authors have read and agreed to the published version of the manuscript.

**Funding:** This research received no external funding.

**Acknowledgments:** We thank Giuseppe Prenesti, a third-year student of Chemical Engineering of Unical (Calabria University), for his help in the calculation of the statistical analysis of this manuscript.

**Conflicts of Interest:** The authors declare that they have no conflict of interest.

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
