# Peer review of "Efficacy of Rapid Maxillary Expansion with or without Previous Adenotonsillectomy for Pediatric Obstructive Sleep Apnea Syndrome Based on Polysomnographic Data: A Systematic Review and Meta-Analysis"

_applsci, doi:10.3390/app10186485_

Round 1
Reviewer 1 Report
Overall this is a potentially worthwhile systematic review on the treatment effects of rapid maxillary expansion on obstructive sleep apnea in children. It does, however, need additional work on the meta-analysis and greater clarity throughout. The more substantive items are outlined here.
METHODS
- The statistical methods involved in the meta-analysis needs significant improvement (p. 4). It is unclear what was done.
- Does “M” refer to “mean”? “Average” is a non-specific lay term that is inappropriate here.
- Rationale for using fixed effects vs random effects model is not given; fixed effects are reported in Fig. 1, yet random effects model calculations are described on 8-9.
- Explain how the observed high heterogeneity (I-squared = 97%) impacted your choice of model and analysis.
- For improvement of the presentation of the methodology and the results, see Camacho et al 2015, SLEEP 385): 669-675 as a template.
- “However, 48 articles were also included to explore and to have a better clinical diagnosis and treatment of childhood OSAS, due to scarcity of randomized clinical trials on this topic.” p. 5. As written, this statement is contrary to the spirit of systematic reviews and evidence-based practice. Re-word: “Due to scarcity of randomized clinical trials on the treatment of childhood OSAS, 48 articles were also included to qualitatively explore this lower quality evidence regarding treatment.”
- It is conventional in a systematic review to summarize these other studies in a tabular form. 3.
- The numbers of studies do not seem to add up: 48 qual. + 6 quant. = 54 total, not 51 as shown in Figure 2. Please clarify. Authors may wish to seek more expertise on meta-analysis, a highly specialized type of analysis, as well as use software for this purpose such as REVMAN (free) to perform the analysis.
DISCUSSION
- Did you mean so say: “This confirms the [importance of ??] interdisciplinary approaches on the treatment of children with OSAS, since both RME and AT are needed treatments…” (p. 11)
- “The authors reported that she had tonsil hypertrophy, which may confirm the secondary outcome of this meta-analysis.” Change “may confirm to “is congruent with”.
- “A significant decrease in oral breathing and improved nasal respiration was described in most of the articles of this meta-analysis, demonstrating the efficacy of RME in contributing to the treatment of OSAS.” How many of the 51 (or 6?) articles? This statement is troubling (“most of the articles”) in that it is not in the quantitative spirit of a systematic review. The authors must scrutinize these articles and report their specific claims regarding oral vs nasal breathing, AND whether or not these modes of respiration were assessed objectively or subjectively (see below); a table reporting the NUMBERS of articles making such claims should be provided.
- Just below this, the strong statements regarding oral vs nasal breathing require a careful re-examination of the original sources (13, 37) as they do not appear to have made objective measurements of these breathing modalities, but instead have relied on subjective reports of parents. A careful consideration of evidence concerning the effects of RPE on nasal vs oral breathing is vital; it remains unclear whether RPE just enlarges the bony framework that defines the nasal passages, or additionally, enlarges the oropharynx (soft tissue). Please re-write this section accordingly. In making conclusions, limit yourself to objective findings of the articles reviewed, not their conjecture.
- “Mandibular advancement devices” is used multiple times; this section requires elimination of redundancies; needs to be tightened. (p. 12)
- “The AHI before and after OMT reported by reviewed authors decreased… (p.12). By “by reviewed authors”, do you mean “in the articles reviewed here”?
- OMT is based on exercises and other strategies that favor sensitivity” Replace “favors” with “are intended to improve”.
Reviewer 2 Report
This review article was well-organaized written. The conclusion was leaded by a meta-analysis using six literatures.
I just concern about the bias of choice of literatures. One of them is a case report. The others are included the literature from same authors. In discussion part, the authors had better to mention to this.
Reviewer 3 Report
This was interesting article. Authors demonstrated the beneficial role of RME for the treatment of OSA with or without AT via literature reviews. As RPE may reduce the resistance of airflow in the nasal cavity, it would be helpful for some cases of peripheral type OSA. But it will not be effective for the treatment of central type OSA. It should be discussed to prevent misleading readers. There were some minor points to be corrected.
- The studies that did not discuss RME had been excluded. However, the studies of RME might be positive attitude for RME in OSA treatment. It could be selection bias. Therefore, neutral or negative result of RME in OSA treatment should be discussed in Discussion section. They might not be included in the analysis if they did not meet the criteria, but should be discussed.
- Generally, abbreviation is not used for title.
Round 2
Reviewer 1 Report
Respond to author
In the manuscript with line numbers the corrections (2 instances) are in line number 173: change "patients' number" to "number of patients".Author Response
In the manuscript with line numbers the corrections (2 instances) are in line number 173: change "patients' number" to "number of patients".
Thank you for noticing this, we have changed it in this way:
The weighted mean was calculated using only one weight (number of patients) for AHI change % and two weights (number of patients and SD) for the other parameters. The change was highlighted in red.Reviewer 3 Report
This manuscript has been successfully revised.
Author Response
This manuscript has been successfully revised.
Thank you for your work